# Evolving Paradigms in the Systemic Treatment of Advanced Gallbladder Cancer: Updates in Year 2022

**DOI:** 10.3390/cancers14051249

**Published:** 2022-02-28

**Authors:** Zishuo Ian Hu, Kian-Huat Lim

**Affiliations:** Division of Oncology, Department of Internal Medicine, Barnes-Jewish Hospital and The Alvin J. Siteman Comprehensive Cancer Center, Washington University School of Medicine, St. Louis, MO 63110, USA; huzi@wustl.edu

**Keywords:** gallbladder cancer, GemCis, HER2, FGFR, PD-L1, immunotherapy

## Abstract

**Simple Summary:**

Gallbladder cancer is distinct type of biliart tract cancer that is rare, aggressive and with limited treatment options aside from surgical resection. As of now in year 2022, systemic chemotherapy remains as the mainstay treatment option for patients with advanced staged gallbladder cancer. Despite decades of scientific research, new treatment options have been struggling to succeed. Furthermore, almost all clinical studies on gallbladder cancer have included other tyes of biliary tract cancers, raising the need to specifically inspect the outcomes of these clinical trial regimens on gallbladder cancer. In this article, we summarized all seminal literature and the most recent advances in scientific discoveries and clinical trials on gallbladder cancer. We provide a succinct update on current understanding, treatment landscape and therapeutic challenges in gallbladder cancer, as well as future prospects in the management of this disease.

**Abstract:**

Gallbladder cancer (GBC) is a biological, anatomical, and clinically distinct subset of biliary tract cancers (BTC), which also include extra- and intra-hepatic cholangiocarcinoma. The advent of next-generation sequencing (NGS) clearly shows that GBC is genetically different from cholangiocarcinoma. Although GBC is a relatively rare cancer, it is highly aggressive and carries a grave prognosis. To date, complete surgical resection remains the only path for cure but is limited to patients with early-stage disease. The majority of the patients are diagnosed at an advanced, inoperable stage when systemic treatment is administered as an attempt to enable surgery or for palliation. Gemcitabine and platinum-based chemotherapies have been the main treatment modality for unresectable, locally advanced, and metastatic gallbladder cancer. However, over the past decade, the treatment paradigm has evolved. These include the introduction of newer chemotherapeutic strategies after progression on frontline chemotherapy, incorporation of targeted therapeutics towards driver mutations of genes including HER2, FGFR, BRAF, as well as approaches to unleash host anti-tumor immunity using immune checkpoint inhibitors. Notably, due to the rarity of BTC in general, most clinical trials included both GBC and cholangiocarcinomas. Here, we provide a review on the pathogenesis of GBC, past and current systemic treatment options focusing specifically on GBC, clinical trials tailored towards its genetic mutations, and emerging treatment strategies based on promising recent clinical studies.

## 1. Introduction

Gallbladder cancer (GBC) is a rare but deadly malignancy, with an estimated 5-year survival rate of 2% in metastatic disease. Prognosis is particularly poor in older patients and racial minorities [1]. Worldwide, the incidence of GBC varies significantly based on geographic location. Incidence rates are very high in South America, high in Japan and Korea, moderate in Eastern and Central Europe, and low in North America. In the United States, GBC is a relatively rare cancer. In 2022, an estimated 12,130 GBC and other biliary cancers are expected to be diagnosed, and 4400 will die from these diseases [2]. Here we review the risk factors for developing GBC, its pathophysiology, current systemic treatments, and ongoing clinical trials with a focus on targeted therapy for GBCs. 

## 2. Risk Factors

In addition to ethnicity, a host of other risk factors have been implicated in the development of GBC, including existing cholelithiasis, gender, older age, obesity, occupation, *Salmonella* infection, and genetic predisposition. In the United States, American Indian and Hispanic women have the highest incidence and mortality rates of GBC [3]. In more recent years, there has also been an increase in GBC incidence rates in the African American population [4]. Gallstones are found in 69–86% of all GBC patients and have been strongly associated with GBC [5,6,7,8]. However, the incidence of GBC in patients with gallstones was reported to be 0.5, and the majority of patients with gallstones will not develop gallbladder cancer in their lifetimes [9]. Older age is associated with GBC, with a median age of 67–72 years [10,11,12]. Women are two to six times more likely to develop GBC than men [11]. Obesity and occupation in the textile, chemical processing, and petroleum refining industries have also been linked to GBC [13,14,15,16,17]. In *Salmonella*-endemic regions of the world, such as the Indian subcontinent and central and South America, the chronic infection has been linked to GBC [18,19]. Patients with Lynch syndrome were also reported to be at a higher risk of developing GBC [20].

## 3. Pathogenesis

The pathogenesis of GBC is thought to derive largely from chronic inflammation. The current gallbladder carcinogenesis model postulates that chronic inflammation of the gallbladder leads to hyperplasia and metaplasia, which can develop into dysplasia, then in situ carcinoma, and eventually invasive carcinoma in a step-wise fashion [21,22]. Roa et al., used mapping techniques on gallbladders with invasive cancers and found that metaplasia, dysplasia, and carcinoma in situ were adjacent to cancer in 66%, 81.3%, and 69% of the cases, respectively [23,24].

The exact etiology of chronic inflammation of the gallbladder varies by ethnicity. Chronic inflammation brought on by gallstones promotes a pro-carcinogenic microenvironment, stimulating tumor proliferation and progression [25]. GBC in South America has largely been attributed to chronic inflammation from gallstones. Twenty-seven percent of the adult population in Chile are estimated to have gallstones, and Chile has one of the highest rates of GBCs in the world, with an estimated 9.2 cases per 100,000 [26,27]. Indigenous Chilean Mapuche Indian women, in particular, have very high rates of gallstone prevalence and mortality [28,29]. The Chile Biliary Longitudinal Study (Chile BiLS) is a study of 4726 Chilean women with gallstones enrolled from 2016 to 2019 that are being followed for gallbladder dysplasia or cancer for six years [30]. Analysis of the inflammatory profile of 200 Mapuche women with gallstones in the BiLS study found that they expressed higher levels of the inflammatory cytokine IL-8 compared to 200 non-Mapuche women with gallstones, suggesting that there may be ethnic differences in the inflammatory response to gallstones [31].

In Japan and China, a series of studies also reported an association between GBCs and anomalous pancreaticobiliary ductal junction (APDJ), a rare congenital anomaly [32,33,34,35,36]. Patients with APDJs experience chronic reflux of pancreatic juice into the bile duct, causing chronic inflammation in the biliary tract, which can eventually lead to hyperplasia, metaplasia, and cancer of the bile ducts. A meta-analysis of nine case-control studies found that the incidence of APDJ was higher in GBC patients than in control patients (10.60% vs. 1.76%, OR: 7.41, 95% CI: 5.03 to 10.87, *p* < 0.00001) [37].

## 4. Genomics

Genomic sequencing has revealed significant heterogeneity in genes commonly mutated in biliary tract cancers (BTCs). Mutations in *FGFR1*, *FGFR2*, *IDH1*, *IDH2*, *BAP1*, and *ARID1A* are more commonly found in intrahepatic cholangiocarcinoma, while *SMAD4* mutations are more commonly seen in extrahepatic cholangiocarcinomas [38,39,40,41,42].

GBCs are molecularly distinct from other BTCs. A number of studies have reported increased activation of the *EGFR* family of genes in GBCs [42,43,44]. Other genetic mutations found to be associated with GBC include *TP53*, *SMAD4*, *ARID1A*, *PIK3CA*, *CDKN2A*, and *CDKN2B* [42,44,45,46,47]. Weinberg et al., reviewed 1502 BTCs using next-generation sequencing (NGS), immunohistochemistry, in situ hybridization, and RNA sequencing and found that GBCs had significantly higher Her2/neu overexpression (9.2%) and amplification (1.44%) and high TOP2A expression (78.3%) and amplification (25%) compared to cholangiocarcinomas [44]. The authors also found that 19.1% of 428 GBCs had mutations in homologous repair genes.

There was also some evidence that GBC tumor mutations vary depending on geographic location [48]. Narayan et al., reviewed 81 patients globally, with 21 patients from Chile and 11 patients from Japan [48]. They found that several mutations were notably absent in specific populations. *ARID1A* and *PIK3CA* mutations were not present in the Japanese cohort. *ARID2* and *ERBB3* mutations were absent in Chilean patients.

It is unclear whether HER2/neu overexpression is associated with worse survival in GBCs. Based on a retrospective evaluation by Vivaldi et al., HER2 overexpression was associated with lower 5-year overall survival (OS, 34% vs. 41%) and shorter disease-free survival (DFS, 10.6 vs. 20.9 months) compared to HER2-negative BTCs [49]. Although it is difficult to extrapolate this result to GBCs in particular since 12 of the 13 GBC patients in the study were HER2 negative. Roa et al., performed immunohistochemistry in 187 cases of GBCs and also found that HER2/neu was overexpressed in 12.8% of cases but did not find a statistically significant difference in OS at 5 years between HER2 positive and HER2 negative cases [43]. The molecular pathogenesis of GBCs may also be attributed to two different genetic pathways. In GBCs arising from APDJs, *KRAS* mutations are common, and *TP53* mutations are relatively late onset [50,51,52]. In GBCs of Chilean patients with cholelithiasis, *KRAS* mutations are rare, and *TP53* mutations are more common [53,54].

## 5. Treatment Options

Developing effective treatment options for GBC has been relatively challenging, given the rarity and aggressiveness of the disease. As a result, despite having different risk factors and profiles, all BTCs, including intrahepatic cholangiocarcinoma, extrahepatic cholangiocarcinoma, and GBC, were often grouped together in large phase 3 clinical trials.

### 5.1. Neoadjuvant Therapy for BTCs

There is currently no preferred neoadjuvant therapy for gallbladder cancer. There is relatively limited data to support a standard regimen or definitive benefit currently. A retrospective review of 74 GBC patients by Creasy et al., found a subset of patients that responded to neoadjuvant therapy and had improved outcomes after definitive surgery [55]. Shroff et al., evaluated the use of gemcitabine, cisplatin, and nab-paclitaxel in 60 patients with advanced BTCs in a phase 2 study [56]. Thirteen of the patients had GBC. ORR was 45%, and the disease control rate was 84%. Notably, 12 patients were converted from unresectable to resectable disease and underwent surgery. Other chemotherapy options for neoadjuvant therapy include single-agent 5-FU, single-agent capecitabine, single-agent gemcitabine, a gemcitabine–platinum combination, or a 5-FU–platinum combination. 

### 5.2. Adjuvant Therapy

Surgery remains the only curative option for GBC. Despite curative resection, 66% of patients with GBC were reported to develop disease recurrence within 2 years, often at a distant site [57]. Although there is currently no established neoadjuvant therapy for BTCs, a few clinical trials have been reported recently for adjuvant therapy in BTCs (Table 1). 

The BILCAP study was a phase 3, randomized multi-institutional study conducted in the United Kingdom that compared capecitabine to observation in patients that had cholangiocarcinoma or GBC who had undergone a macroscopically complete resection with curative intent [58]. Four hundred and forty-seven patients were enrolled over the course of 2006 and 2014. Seventy-nine GBC patients were in the study. There was no statistically significant improvement in OS with the capecitabine group in the intention-to-treat (ITT) analysis (median 51.1 versus 36.4 months, hazard ratio 0.81, 95% CI 0.63–1.04; *p* = 0.097). However, after sensitivity analysis in the ITT population, which adjusted for minimization factors, nodal status, grade, and gender, the HR for OS was 0.71 (95% CI, 0.55–0.92; *p* = 0.01). Based on the findings of this study, current NCCN guidelines recommend 6 months of adjuvant capecitabine for resected GBC.

The PRODIGE-12/ACCORD-18 study was a phase 3, randomized multi-institutional French study that compared gemcitabine and oxaliplatin (GEMOX) to observation in patients with cholangiocarcinoma or GBC who had undergone a macroscopically complete resection with curative intent [59]. They recruited 196 patients from 2006 to 2014. There were 38 GBC patients in the study. After a median follow-up of 46.5 months, the study investigators found no difference in median OS between the GEMOX group vs. the observation group (75.8 months vs. 50.8 months, HR 0.71, 95% CI 0.70–1.66, *p* = 0.74). Planned subgroup analyses of the GBCs, ECC, and ICC did not suggest any subgroup that benefited from adjuvant GEMOX. There was actually worse recurrence-free survival (RFS, *p* = 0.034) and OS (*p* = 0.017) in GBC patients that received GEMOX.

SWOG S0809 is a phase II trial that assessed adjuvant capecitabine and gemcitabine followed by concurrent capecitabine-based chemoradiation after resection of ECC and GBC. A total of 79 patients (68% ECC and 32% GBC) were treated. For all patients, 2-year survival for all patients was 65% (95% CI 53% to 74%; 68% for ECC and 56% for GBC, *p* = 0.87). Median OS was 35 months and did not differ between R0 and R1 resections. Interestingly, the first relapse occurred predominantly at distant sites in GBC patients, potentially suggesting the effectiveness of local control for a combined-modality regimen.

### 5.3. Systemic Therapy

Cytotoxic chemotherapy remains the mainstay treatment for patients with inoperable GBC (Table 2). The ABC-02 trial was a phase 3, randomized study that compared gemcitabine plus cisplatin (GemCis) to gemcitabine alone in 410 patients with locally advanced or metastatic cholangiocarcinoma, gallbladder cancer, or ampullary cancer [61]. Median OS was 11.7 months in the GemCis group and 8.1 months in the gemcitabine group (HR 0.64, 95% CI 0.52 to 0.80; *p* < 0.001). PFS was 8.0 months in the GemCis group and 5.0 months in the gemcitabine group (HR 0.63, 95% CI 0.51 to 0.77; *p* < 0.00). Tumor control was 81.4% in GemCis group and 71.8% in the gemcitabine group (*p* = 0.049). Importantly, subgroup analysis showed that the benefit of GemCis was comparable among patients with GBC, ICC, or ECC (HR of 0.61, 0.57, and 0.73, respectively). Based on the results of this study, gemcitabine with cisplatin has been the first-line therapy for locally advanced and metastatic BTC since at least the year 2010.

Building on this regimen, a phase 2 trial that included 60 patients, nab-paclitaxel plus GemCis, resulted in a median PFS of 11.8 months and a median OS of 19.2 months in an intention-to-treat analysis. The partial response rate was 45%, and the disease control rate was 84% [56]. However, due to significant hematologic toxicities from the first 32 enrolled patients, the starting dose of each agent had to be reduced in the remaining patients. A phase III randomized trial of gemcitabine, cisplatin, and nab-paclitaxel versus gemcitabine and cisplatin in newly diagnosed, advanced BTCs (SWOG S1815, NCT03768414) has recently completed accrual, and the results are eagerly awaited. However, it is likely that the triplet regimen will be associated with more side effects and should be offered to highly selected patients with good performance status and organ functions. Due to the higher partial response rate, there is currently high interest in testing this chemotherapy triplet as a neoadjuvant downstaging strategy for patients with initially unresectable GBCs or GBC with high-risk features such as a bulky tumor, liver invasion, and multiple enlarged adjacent lymph nodes.

FOLFOX as second-line therapy for BTC was established by the recently reported ABC-06 trial [62]. The ABC-06 trial was a phase 3, randomized study that compared FOLFOX with active symptom control (ASC) to ASC alone in 162 locally advanced or metastatic BTC patients who had progressed on GemCis. Thirty-four GBC patients were included in the study. For all BTC patients, median OS was significantly longer in the FOLFOX group (6.2 months) compared to the ASC alone group (5.3 months) (HR 0.69, 95% CI 0.50–0.97; *p* = 0.031) at 6 months. After 1 year, the OS rate was 25.9% in the FOLFOX group and 11.4% in the ASC alone group. The objective response rate was 5% in the FOLFOX group. However, it is worth noting, in the GBC subgroup specifically, median OS in the FOLFOX group was 5.1 months compared to 4.6 months in the ASC alone group (HR of 0.56, 95% CI 0.27–1.17).

Besides FOLFOX, the combination of 5-FU/leucovorin and liposomal irinotecan as second-line treatment is another option. In a multicenter, open-label, randomized, phase 2b study performed in South Korea (NIFTY), 174 patients were enrolled (88 treated with liposomal irinotecan plus 5-FU/leucovorin, and 86 in the 5-FU/leucovorin group). At a median follow-up of 11.8 months, the median PFS was significantly longer in the liposomal irinotecan plus 5-FU/leucovorin group (7.1 months, 95% CI 3.6–8.8) compared to 5-FU/leucovorin group (1.4 months, 1.2–1.5; hazard ratio 0.56, 95% CI 0.39–0.81; *p* = 0.0019) [63]. As expected, patients treated with liposomal irinotecan plus 5-FU/LV experienced more side effects, including neutropenia and fatigue. A similar study testing liposomal irinotecan plus 5-FU/LV for biliary cancer is being evaluated in a phase 2 study in the US (NAPOLI-2, NCT04005339).
cancers-14-01249-t002_Table 2Table 2Current systemic options for advanced, inoperable GBC.AuthorsPhaseLine of TreatmentTreatmentMedian PFS (Months)Median OS (Months)Valle [61]  ABC-02 study31GEMCIS vs. gemcitabine8 vs. 5 (*p* < 0.001)11.7 vs. 8.1 (*p* < 0.001)Shroff [56]21GEMCIS + nab-paclitaxel11.819.2Williams [64]21Gemcitabine + carboplatin7.810.6Kim [65]31CAPOX vs. GEMOX  (non-inferior study)5.8 vs. 5.310.6 vs. 10.4 (*p* = 0.131)Lamarca [62]  ABC-06 study32FOLFOX vs. symptom control4 vs. N/A6.2 vs. 5.3 (*p* = 0.031)Yoo [63]  NIFTY study225-FU + liposomal irinotecan7.1 vs. 1.4  (*p =* 0.0019)8.6 vs. 5.5 (*p* = 0.035)Abbreviations: GEMCIS = gemcitabine plus cisplatin. GEMOX = gemcitabine plus oxaliplatin. CAPOX = capecitabine plus oxaliplatin. CT = chemotherapy. RT = radiotherapy. CRT = chemoradiotherapy.

## 6. Targeted Therapy

Although cytotoxic chemotherapy currently remains the mainstay for adjuvant and advanced GBC, the advent of next-generation sequencing has revealed a number of driver mutations in GBC, to which targeted approaches were tested and shown promise in selected patients. Here, we discuss results from clinical trials targeting the HER2, FGFR, BRAF/MEK, and PD-1 pathways (Figure 1) [66,67,68,69,70,71]. IDH-1 mutations are extremely rare in GBC and more common in ICC and ECC and will not be discussed here.

### 6.1. HER2/neu Pathway

Mutations or overexpression of HER2/neu are present in 12–15% of GBC [43]. A number of clinical trials were undertaken to explore the use of HER2/neu-directed therapy in BTCs with mixed results. Two phase 2 trials, by Peck et al., and Ramanthan et al., were conducted with lapatinib tosilate in unselected patients with BTCs, which reported a lack of activity [72,73]. Of the eight evaluable patients treated with lapatinib in Peck et al.’s study, there were no objective responses, and no *HER2/neu* somatic mutations or HER2/neu overexpression were found. Ramanthan et al., enrolled five GBC patients and found no objective responses to lapatinib. TreeTopp was a randomized phase 2 study of varlitinib, a small molecule pan-human HER inhibitor, with capecitabine compared to capecitabine alone in second-line advanced or metastatic BTC [74]. One hundred and twenty-seven unselected patients were randomized, including 34 GBC patients. PFS was similar for patients receiving varlitnib with capecitabine compared to capecitabine alone. ORR was 9.4% in the varlitinib with capecitabine group, but not statistically significant from the 4.8% in the capecitabine alone group (*p* = 0.42).

MyPathway was a multicenter, open-label, phase 2a, basket study that evaluated the use of pertuzumab with trastuzumab in patients with previously treated metastatic BTCs with *HER2* amplification or HER2 overexpression [75]. ORR was 23% in 39 enrolled patients. PFS was 4 months, and OS was 10.9 months. Post-hoc, exploratory analysis of the GBC subgroup found an ORR of 31% in 16 patients and an OS of 14.2 months. These results in MyPathway for metastatic, previously treated GBC patients compare favorably against the results from the ABC-06 trial, where OS was 5.1 months. Notably, this regimen is anticipated to be better tolerated than standard GemCis and should be tested in a randomized study in the future for this population.

Zanidatamab is a bispecific HER2-targeted antibody directed against the juxtamembrane domain (ECD4) and the dimerization domain (ECD2) of HER2. Twenty-one BTC patients (12 GBCs) were enrolled in the expansion cohort of a phase 1 study (NCT02892123) enrolling advanced HER2-expressing cancers with progression after standard of care therapy [76]. The objective response rate was 40%, and the duration of response was 7.4 months. Based on these promising results, HERIZON-BTC-01 is an ongoing, multicenter, open-label phase 2 trial to evaluate zanidatamab in advanced BTCs.

Harding et al., recently reported on the results of a phase 2 basket trial targeting HER2-mutated advanced BTCs with neratinib [77]. Twenty-five patients with BTCs were enrolled (10 GBCs). ORR was 16% with an OS of 5.4 months and PFS of 2.8 months. Other ongoing clinical trials include the HERB trial, a multicenter phase 2 study of trastuzumab deruxtecan for HER2-positive unresectable or recurrent biliary cancer [78], and a phase 1/b study of afatinib in combination with capecitabine in patients with refractory solid tumors [79].

Besides overexpression, activating mutations of HER2/neu, particularly at codon 310 from serine to tyrosine or phenylalanine (S310Y/F), are detected in GBCs. In a multi-histology, open-label, phase II ‘basket’ study (SUMMIT, NCT01953926), neratinib 240 mg daily was tested in patients with somatic *HER2* mutations. As of September 2022, 25 patients with *HER2*-mutant BTC have enrolled: gallbladder (40%), intrahepatic (24%), extrahepatic (20%), and ampulla of Vater (16%). Confirmed ORR in these 25 patients was 12% (95% CI 3–31%). Median PFS and OS were 2.8 (95% CI 1.1–3.7) and 5.4 (95% CI 3.7–11.7) months, respectively [77].

Overall, targeting *HER2/neu* is an active field in GBC with multiple available therapeutic agents. Combination strategies with systemic chemotherapy such as GemCis in carefully designed trials will likely be the next stage of the investigation. 

### 6.2. FGFR Pathway 

*FGFR* mutations or fusions are found in about 20% of intrahepatic cholangiocarcinomas but are relatively rare in extrahepatic cholangiocarcinomas and GBCs (~3%) [80]. One current trial investigating FGFR inhibitors in GBC is PUMCH (NCT04211168), a phase 2, single-center trial in China that is testing the use of toripalimab, a PD-1 inhibitor, with lenvatinib, a multi-kinase inhibitor of FGFR, VEGFR, and PDGFR as second-line treatment in BTCs. NCT04742959 is another a phase I clinical trial testing the use of TT-00420, a spectrum-selective multi-kinase inhibitor in patients with advanced solid tumors [81].

### 6.3. BRAF/MEK Pathway

The Raf/MEK/ERK pathway was reported to be activated in BTCs, with *BRAF V600E* mutations reported in up to 3% of BTCs [82]. The phase 1b ABC-04 trial examined the combination of selumetinib, a MEK inhibitor, in combination with GemCis in advanced BTCs [83]. They found that two out of eight evaluable patients had a partial or complete response. Binimetinib, another MEK inhibitor, was also tested in combination with GemCis and with capecitabine for patients with BTCs in small phase 1/2 trials [84,85]. Thirty-four patients with BTC that progressed on gemcitabine-based first-line therapy received binimetinib and capecitabine and had an ORR of 20.6% with a PFS of 4.1 months and OS of 7.8 months. Thirty-five treatment-naïve BTC patients that received binimetinib with GemCis had an ORR of 36%, PFS of 6 months, and OS of 13.3 months. The ROAR basket trial is a phase 2, open-label trial enrolling patients with recurrent or progressive *BRAF V600E* mutated cancers [86,87]. Forty-three patients with BTCs received dabrafenib, a BRAF inhibitor, and trametinib, a MEK inhibitor. The majority of patients had intrahepatic bile duct cancer, but one patient had GBC. ORR was 47% (95% CI, 31–62) and PFS was 9 months (95% CI, 5–10), and median OS was 14 months (95% CI, 10–33).

### 6.4. PD-1 Pathway

Immunotherapy for solid tumors has garnered enormous success over the past decade, with immune checkpoint inhibitors targeting the PD-1 and CTLA-4 pathways demonstrating improved OS and response rates compared to standard treatments. Pembrolizumab and nivolumab are anti-PD-1 monoclonal antibodies that have shown activity in tumors with mismatch repair deficiency (MMR-D), high tumor mutation burden (TMB), or PD-L1 expressing tumors. MMR-D (~3%) and high TMB cases are rare in BTCs [88], potentially explaining why monotherapy with PD-1 inhibitors has resulted in mixed success. 

Piha-Paul presented data from 104 patients with advanced BTCs from the KEYNOTE-158 trial, which treated patients with advanced cancers with pembrolizumab [89]. With a median follow-up of 7.5 months, ORR was 5.8%, with the median duration of response not reached. Median OS was 7.4 months (95% CI, 5.5–9.6), and median PFS was 2.0 months (95% CI, 1.9–2.1).

A phase 2 trial investigated the efficacy of nivolumab in patients with advanced BTC refractory to first-line therapy [90]. Of the 54 enrolled patients, 17 patients were GBCs. Objective response was observed in 2 of 13 (15%) evaluable patients with GBC. 

TOPAZ-1 is a randomized, double-blind placebo-controlled, international phase III study combining durvalumab or placebo with GemCis for patients with treatment-naive advanced BTCs [91]. In this study, patients were treated with GemCis plus durvalumab or placebo every 3 weeks for eight cycles, followed by durvalumab or placebo every 4 weeks until disease progression. At data cutoff in August 2021 and presented in the Annual ASCO GI Cancer Symposium that at data cutoff in August 2021 for interim analysis, 685 pts were randomized to durvalumab + GemCis (*n* = 341) or placebo + GemCis (*n* = 344). The primary objective was met with durvalumab + GemCis significantly improving median OS compared to placebo + GemCis (12.8 vs. 11.5 months, hazard ratio 0.80; 95% confidence interval 0.66–0.97; *p* = 0.021). PFS was also significantly improved with durvalumab vs. placebo (HR, 0.75; 95% CI, 0.64–0.89; *p* = 0.001). ORR was 26.7% with durvalumab and 18.7% with placebo. Grade 3/4 treatment-related adverse events (TRAEs) were comparable between the two arms [92]. Interestingly, separation of both the PFS and OS curves between the two groups appeared to occur after chemotherapy was completed, raising the question of the benefit of durvalumab when chemotherapy was concurrently administered. Subgroup analysis also showed that Asian patients derived the most benefit from the triple combination than non-Asian patients. The triple combination also seemed to mainly benefit patients with ICC and ECC (HR 0.76 for both) as opposed to GBC (HR 0.94). In addition, the triple combination also appeared to be more beneficial for patients who had recurrent rather than initially unresectable disease and locally advanced versus metastatic disease. Interestingly, the benefit of durvalumab was observed regardless of PD-L1 expression by immunohistochemistry (cutoff of 1%) in the tumor, immune cells, or total tumor area. Final publication of this study should be available in year 2022. Nonetheless, this study raised significant interest and questions targeting specifically non-Asian and GBC patients which should be further addressed in future trials.

## 7. Future Directions

Multiple trials are ongoing exploring the use of immunotherapy, targeted therapy, and cytotoxic therapy for BTCs. A list of ongoing clinical trials targeting GBCs internationally is listed in Table 3. However, lessons from the recent exciting trials clearly highlighted two areas that need to be worked on to improve the outcomes of GBC patients. First, genomic analysis and clinical trial experience clearly showed that targeting the HER2 pathway is the leading strategy for GBC. Second, the lack of efficacy of lapatinib or vartilinib, as opposed to pertuzumab with trastuzumab or zanidatamab suggest that additional mechanisms of action, such as antibody-dependent cellular toxicity, immune-mediated killing, or direct cytotoxicity provided by chemotherapy may be needed to augment the effect of HER2 inhibition in order to achieve a meaningful clinical response. To this end, combination strategies or newer, more potent HER2-targeted agents such as antibody-drug conjugates should be tested. The promise of immunotherapy, though slightly disappointing based on the subgroup analysis from the TOPAZ-1 study, should warrant more basic translational research into GBC. In particular, PD-1 is only one of the many checkpoint mechanisms that silence anti-tumor T cells. Other checkpoints such as LAG3, TIM3, and TIGIT have all been implicated in T cell exhaustion and remain to be targeted. Furthermore, strategies to overcome additional known biological barriers, including tumor desmoplasia and suppressive myeloid cells in the tumor microenvironment, need to be developed and tested in future trial regimens. The positive results of the TOPAZ-1 study, while exciting as this was the first positive trial since ABC-02, are far from being optimal given the mild improvement of median OS from 11.5 to 12.8 months (*p* = 0.021). The plateauing of OS curve after two years for patients treated with durvalumab strongly suggest a need to develop better predictive markers, other than PD-L1 expression by immunohistochemistry, to identify the subgroup of patients who will benefit from checkpoint immunotherapy. 

## 8. Conclusions

Genomic studies have clearly shown that BTCs are a heterogeneous group of cancers biologically and that cholangiocarcinomas and GBC have distinct genetic alterations. In recent years, more and more actionable mutations are being treated for BTCs as a group, but success in GBC remains limited, as clearly discerned by subgroup analysis of these clinical trials. As of now, in the year 2022, combination chemotherapies continue to be the backbone of treatment for most patients in the frontline, second line, and adjuvant settings. The promising response rate and survival data reported from the phase II study employing GEMCIS plus nab-paclitaxel suggest that imposing extensive DNA damage in GBC cancer cells is a feasible strategy [56]. However, systemic toxicities and durability of this strategy are major concerns. Therefore, understanding the mechanisms of resistance to these regimens and developing novel therapeutic combinations to overcome these mechanisms are critical and most likely to make the largest impact for GBC patients. Another important consideration other than the intrinsic genomic alteration of GBC is the ethnicity of the patient as a biological variable that could potentially impact the natural course and treatment response, as discerned from the TOPAZ-1 study. The molecular underpinnings of this difference remain poorly understood and require intensive research in order to aid patient selection in future clinical trials. The development of robust, dedicated preclinical models such as patient-derived xenografts from different ethnic backgrounds and gender, as well as new genetic mouse models to study GBC as a separate disease, are critically needed to develop novel therapeutic strategies that can be further tested in clinical trials. In the meantime, we should also be acutely aware that BTC is an overall rare cancer, and further dissecting it into different histologic and molecular subgroups, including GBC on its own as separate clinical trials, will be challenging. Thus far, all the seminal trials in BTCs have resulted from close international collaborations, which will remain the bedrock for future efforts in this disease. In summary, GBC is a biologically distinct, rare entity within the spectrum of BTCs with limited treatment options and thus requires intensive and collaborative preclinical and clinical efforts to improve its outcome.

## Figures and Tables

**Figure 1 cancers-14-01249-f001:**
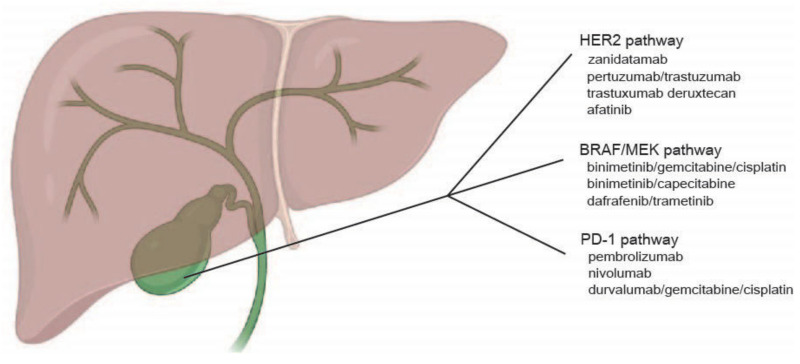
Current therapeutic targets in gallbladder cancer.

**Table 1 cancers-14-01249-t001:** Current adjuvant options after surgical resection of GBC.

Authors	Phase	Treatment	Median RFS (Months)	Median OS (Months)
Primrose [58] BILCAP study	3	capecitabine vs. observation	24.4 vs. 17.5 (*p* = 0.03)	51.1 vs. 36.4 (*p* = 0.097)
Edeline [59] PRODIGE 12-ACCORD 18 study	3	GEMOX vs. observation	30.4 vs. 18.5 (*p* = 0.48)	75.8 vs. 50.8 (*p* = 0.74)
Ben-Josef [60] SWOG S0809 study	2	Gemcitabine and capecitabine followed by capecitabine with RT	26	35

**Table 3 cancers-14-01249-t003:** Ongoing clinical trials for GBC.

Phase	NCT Number	Tumor Type	Line	Treatment	Location
2/3	NCT02867865	Locally advanced gallbladder cancers	1st	Neoadjuvant GEMCIS vs. neoadjuvant radiation with weekly gemcitabine	India
2/3	NCT04559139	Stage II-III gallbladder cancers	1st	Neoadjuvant GEMCIS, resection then adjuvant GEMCIS vs. resection then adjuvant GEMCIS	United States
2	NCT04333927	Resected extrahepatic cholangiocarcinoma and gallbladder cancers	1st	Camrelizumab, then capecitabine with radiotherapy vs. observation	China
1	NCT03257761	Unresectable liver, pancreatic, BTCs	2nd	Guadecitabine and durvalumab	United States
3	NCT03673072	BTCs	1st	Neoadjuvant GEMCIS followed by liver resection vs. upfront liver resection	Germany
2	NCT03833661	Locally advanced BTCs	2nd	bintrafusp alfa, a bifunctional anti-PD-L1/TGFβ trap	United States, Europe, Asia
2	NCT03473574	Unresectable BTCs	1st	durvalumab/tremelimumab/gemcitabine vs. durvalumab/tremeliumab/GEMCIS vs. GEMCIS vs. durvalumab/GEMCIS	Germany
2	NCT03043547	Locally advanced BTCs	2nd	Liposomal irinotecan and 5-FU vs. 5-FU	Germany
3	NCT02170090	Resected BTCs	1st	Adjuvant GEMCIS vs. capecitabine	Europe, Australia
2	NCT04466891	Locally advanced HER2-amplified BTCs	2nd	Zanidatamab, a HER2-targeted bispecific antibody	United States, Europe, Asia
2/3	NCT04066491	Locally advanced BTCs	1st	bintrafusp alfa with GEMCIS vs. GEMCIS	United States, South America, Australia, Asia, Europe
3	NCT03779035	Resected BTCs	1st	Adjuvant GEMCIS vs. capecitabine	China
1/2	NCT04203160	Locally advanced BTCs	1st	Devimistat, anti-mitochondrial inhibitor, with GEMCIS vs. GEMCIS	United States
2	NCT04308174	BTCs	1st	Neoadjuvant durvalumab with GEMCIS vs. neoadjuvant GEMCIS	Korea
2	NCT02151084	Locally advanced or metastatic BTCs	1st	Selumetinib with GEMCIS vs. GEMCIS	Canada
2	NCT02834013	Locally advanced gallbladder cancers	2nd	Nivolumab and ipilimumab	United States
2	NCT03260712	Locally advanced or metastatic BTCs	1st	Pembrolizumab with GEMCIS	Europe
2	NCT03801083	Locally advanced or metastatic BTCs	1st and 2nd	Tumor infiltrating lymphocytes and IL-2	United States
1/2	NCT03733990	Locally advanced or metastatic BTCs, melanoma, ER+ breast, gastric, ovarian, pancreatic, colorectal, liver or anaplastic thyroid cancers	2nd	FP-1305, a CLEVER-1 inhibitor	United States, Europe
2	NCT04856761	Resected BTCs	1st	Adjuvant capecitabine vs. adjuvant S1	China
1	NCT04495296	Metastatic solid tumors	2nd	TST001, an anti-Claudin 18.2 monoclonal antibody	China
2	NCT03796429	Locally advanced BTCs	1st	Toripalimab with GEMCIS	China
2	NCT04059562	Locally advanced or metastatic BTCs	2nd	Trifluridine/tipiracil with irinotecan	Germany
2	NCT04969887	Intrahepatic cholangiocarcinomas and gallbladder cancers	1st and 2nd	Ipilimumab and nivolumab	Australia
1/2	NCT05000294	Metastatic BTCs	2nd	Atezolizumab with tivozanib	United States
2	NCT03278106	Advanced BTCs	2nd	Trifluridine/tipiracil	United States
3	NCT03768414	Metastatic or locally advanced BTCs	1st	GEMCIS vs. GEMCIS with nab-paclitaxel	United States
1/2	NCT04742959	Metastatic BTCs	2nd	TT-00420, a spectrum-selective multi-kinase inhibitor	United States
2	NCT04383210	NRG1 gene fusion positive advanced BTCs	1st and 2nd	Seribantumab, an anti-Her3 monoclonal antibody	United States
1/2	NCT04426669	Metastatic gastrointestinal epithelial cancers	2nd	CISH inactivated tumor infiltrating lymphocytes and IL-2	United States
2	NCT04941287	Unresectable BTCs	2nd	Atezolizumab with varlilumab, an anti-CD27 antibody vs. atezolizumab with varlilumab and cobimetinib	United States
1/2	NCT05086692	Advanced solid tumors	Any	MDNA11, an engineered IL-2	Australia
1/2	NCT04430738	HER2-positive GI cancers	1st	Tucatinib with trastuzumab and FOLFOX vs. tucatinib with trastuzumab and CAPOX	United States
3	NCT04924062 and NCT04924062	Advanced or unresectable BTCs	1st	Pembrolizumab with GEMCIS vs. GEMCIS	Global
2	NCT04211168	Advanced BTCs	2nd	Toripalimab with lenvatinib	China
2	NCT02703714	Advanced BTCs	Any	Pembrolizumab with G-CSF	United States
1	NCT03985072	Advanced solid tumors	2nd	ANDES-1537, an antisense oligonucleotide	Chile
1	NCT04853017	KRAS mutated solid tumor	Any	ELI-002 2P, mix of modified KRAS peptides	United States
1/2	NCT04068194	Advanced or metastatic Hepatobiliary malignancies	2nd	RT with avelumab vs. RT with avelumab and peposertib, a DNA-PK inhibitor	United States
2	NCT02520141	Locally advanced or metastatic BTCs	2nd	Ramucirumab	United States
1	NCT02495896	Advanced solid tumors	Any	sEphB4-HSA fusion protein with gemcitabine and nab-paclitaxel vs. sEphB4-HSA fusion protein with docetaxel vs. sEphB4-HSA fusion protein with GEMCIS	United States

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
