# Peer review of "Evolving Paradigms in the Systemic Treatment of Advanced Gallbladder Cancer: Updates in Year 2022"

_cancers, 2022, doi:10.3390/cancers14051249_

Round 1
Reviewer 1 Report
The paper addresses a very important issue, namely the systemic treatment of advanced gallbladder cancer. This pathology has a low incidence, and the clinical studies presented in recent years are few and have many limitations.
That is why the analysis of the specialized literature presented in this paper is of special interest. The quality of the article is very good, the issues are very clearly and coherently presented.
The presentation of some aspects regarding the genomic sequencing and the exhaustive analysis of the treatment options increase the value of the paper. The current specialty literature is discussed.
In view of the purpose of the study, I consider that the conclusions are not clear.
The conclusions describe some general aspects. I recommend rewording this section.
Author Response
We thank Reviewer 1’s positive comments. As suggested, we have reworded the conclusion (Line 380).
Reviewer 2 Report
Hu and Lim provide a very exhaustive review on the systemic treatment for advanced gallbladder cancer. There are only few minor comments
The FGFR pathway is rarely involved in the gallbladder cancer as the authors also stipulated therefore that section might be reduced to one or two sentences.
The authors should be carefull for the abreviations that are not explained in the text starting with BTC in the abstract ,probably billiary tract cancers
The authors present in their paper mainly trials related to immuno and genic therapy. None of those trials had a conclusive results but they might represent path fofuture researches. Therefore the review might be taken into consideration for publication.
Author Response
We thank Reviewer 2’s recommendations and have reduced the FGFR pathway section to two sentences. We have corrected the BTC abbreviation.
Reviewer 3 Report
This is a well-written review on GBC. The manuscript needs a few improvements:
- A table summarizing the systemic therapies.
- A table summarizing the evidence, including the retrospective data, that adjuvant chemotherapy has a role in resected GBC.
- One paragraph discussing neoadjuvant therapy.
- A few minor corrections, like : next general sequencing--> next-generation sequencing. Please spell out BTC in the abstract.
Author Response
We thank Reviewer 3’s recommendations and spelled out BTC in the abstract and changed next general sequencing to next-generation sequencing. We have also included:
- A table summarizing the systemic therapies (Table 2)
- A table summarizing the evidence including retrospective data for adjuvant chemotherapy in resected GBC (Table 1)
- A paragraph discussing neoadjuvant therapy (line 144)
Round 2
Reviewer 3 Report
Critique has been adequately addressed.